# Melatonin and Its Metabolites Can Serve as Agonists on the Aryl Hydrocarbon Receptor and Peroxisome Proliferator-Activated Receptor Gamma

**DOI:** 10.3390/ijms242015496

**Published:** 2023-10-23

**Authors:** Andrzej T. Slominski, Tae-Kang Kim, Radomir M. Slominski, Yuwei Song, Shariq Qayyum, Wojciech Placha, Zorica Janjetovic, Konrad Kleszczyński, Venkatram Atigadda, Yuhua Song, Chander Raman, Cornelis J. Elferink, Judith Varady Hobrath, Anton M. Jetten, Russel J. Reiter

**Affiliations:** 1Department of Dermatology, University of Alabama at Birmingham, Birmingham, AL 35294, USA; tkim@uabmc.edu (T.-K.K.); yuweisong@uabmc.edu (Y.S.); sqhusain@gmail.com (S.Q.); zjanjetovic@uabmc.edu (Z.J.); venkatra@uab.edu (V.A.); chanderraman@uabmc.edu (C.R.); 2Department of Genetics, University of Alabama at Birmingham, Birmingham, AL 35294, USA; rslominski@uabmc.edu; 3Department of Biomedical Informatics and Data Science, University of Alabama at Birmingham, Birmingham, AL 35294, USA; 4Brigham’s Women’s Hospital, Harvard University, Boston, MA 02115, USA; 5Department of Medicinal Biochemistry, Collegium Medicum, Jagiellonian University, 31-008 Kraków, Poland; wojciech.placha@uj.edu.pl; 6Department of Dermatology, University of Münster, Von-Esmarch-Str. 58, 48161 Münster, Germany; konrad.kleszczynski@ukmuenster.de; 7Department of Biomedical Engineering, University of Alabama at Birmingham, Birmingham, AL 35294, USA; yhsong@uab.edu; 8Department of Pharmacology and Toxicology, University of Texas Medical Branch, Galveston, TX 79567, USA; coelferi@utmb.edu; 9Sygnature Discovery, Discovery Building, Biocity, Nottingham NG1 1GF, UK; j.hobrath@sygnaturediscovery.com; 10Cell Biology Section, NIEHS, National Institutes of Health, Research Triangle Park, NC 27709, USA; jetten@niehs.nih.gov; 11Department of Cell Systems and Anatomy, UT Health, Long School of Medicine, San Antonio, TX 78229, USA; reiter@uthscsa.edu

**Keywords:** melatonin, melatonin metabolites, AhR, PPARγ, receptors, modeling

## Abstract

Melatonin is widely present in Nature. It has pleiotropic activities, in part mediated by interactions with high-affinity G-protein-coupled melatonin type 1 and 2 (MT1 and MT2) receptors or under extreme conditions, e.g., ischemia/reperfusion. In pharmacological concentrations, it is given to counteract the massive damage caused by MT1- and MT2-independent mechanisms. The aryl hydrocarbon receptor (AhR) is a perfect candidate for mediating the latter effects because melatonin has structural similarity to its natural ligands, including tryptophan metabolites and indolic compounds. Using a cell-based Human AhR Reporter Assay System, we demonstrated that melatonin and its indolic and kynuric metabolites act as agonists on the AhR with EC_50_’s between 10^−4^ and 10^−6^ M. This was further validated via the stimulation of the transcriptional activation of the *CYP1A1* promoter. Furthermore, melatonin and its metabolites stimulated AhR translocation from the cytoplasm to the nucleus in human keratinocytes, as demonstrated by ImageStream II cytometry and Western blot (WB) analyses of cytoplasmic and nuclear fractions of human keratinocytes. These functional analyses are supported by in silico analyses. We also investigated the peroxisome proliferator-activated receptor (PPAR)γ as a potential target for melatonin and metabolites bioregulation. The binding studies using a TR-TFRET kit to assay the interaction of the ligand with the ligand-binding domain (LBD) of the PPARγ showed agonistic activities of melatonin, 6-hydroxymelatonin and *N*-acetyl-*N*-formyl-5-methoxykynuramine with EC_50_’s in the 10^−4^ M range showing significantly lower affinities that those of rosiglitazone, e.g., a 10^−8^ M range. These interactions were substantiated by stimulation of the luciferase activity of the construct containing PPARE by melatonin and its metabolites at 10^−4^ M. As confirmed by the functional assays, binding mode predictions using a homology model of the AhR and a crystal structure of the PPARγ suggest that melatonin and its metabolites, including 6-hydroxymelatonin, 5-methoxytryptamine and *N*-acetyl-*N*-formyl-5-methoxykynuramine, are excellent candidates to act on the AhR and PPARγ with docking scores comparable to their corresponding natural ligands. Melatonin and its metabolites were modeled into the same ligand-binding pockets (LBDs) as their natural ligands. Thus, functional assays supported by molecular modeling have shown that melatonin and its indolic and kynuric metabolites can act as agonists on the AhR and they can interact with the PPARγ at high concentrations. This provides a mechanistic explanation for previously reported cytoprotective actions of melatonin and its metabolites that require high local concentrations of the ligands to reduce cellular damage under elevated oxidative stress conditions. It also identifies these compounds as therapeutic agents to be used at pharmacological doses in the prevention or therapy of skin diseases.

## 1. Introduction

Melatonin (*N*-acetyl-5-methoxytryptamine) is produced by a wide range of organisms, including bacteria, unicellular and multicellular fungi, plants and animal species, including invertebrates and vertebrates [1,2,3,4,5]. It is a product of multistep tryptophan metabolism via serotonin and *N*-acetylserotonin [1,5,6,7], and it is rapidly metabolized through indolic and kynuric pathways or via nonenzymatic processes induced by UVR or free radicals [8,9,10]. In mammals, the pineal gland is a major source of melatonin, which enters serum and the cerebrospinal fluid of the third ventricle of the brain to regulate the circadian system. It is also synthesized in many extrapineal and extracranial sites, including skin [11,12]. In addition to the regulation of the circadian rhythm, melatonin can act as a neurotransmitter, hormone, immunomodulator and biological-response modifier, and at its physiological concentrations (≤nM) these functions can be mediated by interactions with high-affinity G-protein-coupled melatonin type 1 and type 2 (MT1 and MT2) receptors [13,14,15]. Interacting with membrane-bound MT receptors melatonin inhibits the production of cAMP and cGMP, which modify protein kinase A and C, and CREB (cAMP response element-binding protein) signaling and activate MAP kinases [13,14,15,16]. MT2 shows a 60% homology in structure to MT1 [17]. In addition to homodimerization, MT1 and MT2 can heterodimerize, which can affect the pharmacological properties of the receptors [18,19], and their activities are also modulated by post-translational modification [16].

Melatonin also stimulates antioxidative responses and DNA repair pathways, which require high local concentrations (1–1000 µM) of the compound and are MT1- and MT2-independent [20,21,22]. These properties are shared by its metabolite products of the indolic and kynuric degradation pathways [8,9,10,21,23,24,25]. Some of them may be in part explained by the binding to quinone reductase 2 (NQO2) [16,26,27], calmodulin (CaM) [28,29,30] or the regulation of mitochondrial functions that affect cellular homeostasis [20,25,31,32]. However, an alternative explanation that melatonin and its metabolites act on nuclear receptors (NR) deserves serious consideration. Since it is now documented that RORα (retinoic acid orphan receptor alpha) is an NR for sterols and secosteroids, but not for melatonin [33,34,35], we have started to search for another NR for melatonin and its metabolites.

The aryl hydrocarbon receptor (AhR) is a perfect candidate for such a receptor, because melatonin and its metabolites show structural similarities to its natural ligands, including tryptophan metabolites and indolic compounds [36,37,38]. The AhR can be activated by structurally diverse synthetic and naturally occurring chemicals, including halogenated aromatic hydrocarbons (HAHs) and nonhalogenated polycyclic aromatic hydrocarbons (PAHs) [39,40]. The AhR contains a promiscuous ligand-binding domain (LBD) accommodating chemicals with distinct structures, such as endogenously produced bilirubin, arachidonic acid and metabolites of lipoxin A4 and prostaglandin G [41] and derivatives of tryptophan, including indigo dye, indole acetic acid and indirubin [36,37,38,42]. Therefore, we performed docking simulations using a homology model of the AhR [43], in conjunction with the binding of melatonin and its indolic and kynuric metabolites to the ligand-binding domain (LBD) of the AhR, transcriptional stimulation of *CYP1A1* promoter and AhR translocation from the cytoplasm to the nucleus in human keratinocytes to assess them as AhR agonists. We have also explored whether melatonin and its metabolites can interact with another nuclear receptor, the peroxisome proliferator-activated receptor gamma (PPARγ).

## 2. Results

The structures of melatonin and its metabolites in comparison to their precursor L-tryptophan and natural ligands of the AhR and PPARγ are presented in Table 1.

### 2.1. Functional Testing of AhR as the Receptor for Melatonin and Its Metabolites

Representative dose-dependent interactions of melatonin and its metabolites 6-hydroxymelatonin (6(OH)melatonin), 5-methoxytryptamine (5-MTT) and *N*-acetyl-*N*-formyl-5-methoxykynuramine (AFMK), products of indolic and kynuric metabolism, respectively, are shown in Figure 1. We used a commercially available cell-based human AhR reporter assay system to analyze the effect of melatonin and its metabolites on AhR-mediated transactivation. There was a marked activation of AhR activity by melatonin with lower potency for 6(OH)melatonin, AFMK and 5-MTT. The corresponding EC_50_ values for melatonin metabolites were, however, higher than that of IAA, a natural ligand for the AhR. These values representing two independent experiments for each compound performed in three or four replicas are shown in Table 2. The interactions of melatonin and its metabolites with the AhR are further substantiated by their ability to stimulate the transcriptional activation of the *CYP1A1* promoter (inserts to Figure 1).

These receptor ligand interactions were further substantiated by data showing that melatonin and its metabolites promoted AhR translocation from the cytoplasm to the nucleus in human keratinocytes, as demonstrated using ImageStream II cytometer analyses (Figure 2A). The left panel shows brightfield, nuclear staining (NS–Hoechst), AHR and merged (NS/AHR) images. In the merged images, there are red stains corresponding to the AhR protein within the nucleus. For each treatment, we analyze the data from close to 1000 individual cells without any selection bias, hence allowing high power for sensitive assessment for changes in expression and/or localization. The right panel of Figure 2A shows the quantification of the translocation as the percentage of AhR staining in the nucleus and as ratios of nuclear (co-localization with Hoechst) vs. cytoplasmic localization of the AhR. The differences are statistically significant, as marked by asterisks. The translocation of the AhR to the nucleus following treatment with melatonin and its metabolites was further confirmed using Western blot (WB) analyses of nuclear and cytoplasmic fractions probed with antibodies against AhR and re-probed with antibodies against proper control markers for cytoplasm and the nucleus (Figure 2B). The left panel shows a representative WB blot indicating AhR translocation in relation to cytoplasmic and nuclear markers, while the right panel represents the quantification of data from three independent experiments with significance indicated by asterisks. For the analysis of whole-cell and cytosolic fractions, band intensities were normalized relative to the loading control and to the band intensities seen in control cells (ethanol), and they are presented as a % of the control (mean ± SD), with statistical significance indicated above the bars. These experimental data provide strong evidence that melatonin and its indolic and kynuric metabolites can act as agonists on the AhR. These findings formed the basis for molecular modeling of their interactions with the AhR.

### 2.2. Interactions of Melatonin and Its Metabolites with PPARγ

The representative dose-dependent bindings of melatonin and its metabolites to the LBD of the PPARγ using the LanthaScreen™ TR-FRET PPARγ Coactivator Assay Kit are shown in Figure 3A. The EC_50_ varied from 10^−4^ to 10^−3^ M, being significantly higher than those for rosiglitazone, e.g., a 10^−8^ M range (Table 2). The EC_50_ values in Table 2 are from two independent experiments with three to four replicas per assay.

Figure 3B shows statistically significant stimulation of the transcriptional activity of the peroxisome proliferator-activated receptor responsive element (PPRE)_3_ containing *ApoA-II* gene promoter induced by 10^−4^ M melatonin, 5-hydroxytryptophan (5(OH)tryptophan), 6(OH)melatonin or AFMK. Pioglitazone and fenofibrate (PPAR agonists; at 2 × 10^−5^ M) were used as positive controls. The *Y*-axis shows relative luminescence units (RLUs) and the *X*-axis shows the control (non-ligand stimulated promoter) and the PPAR activators used. In relation to pioglitazone and fenofibrate, strong activators of PPAR, melatonin and its indole and kynurenic metabolites are able to enhance the PPAR-mediated activation of the PPRE-driven promoter at a concentration of 10^−4^ M, which is 1–2 orders of magnitude higher than the corresponding synthetic analogs (Figure 3B).

### 2.3. Molecular Docking and Binding Thermodynamics Analysis of Melatonin and Its Metabolites Targeting AhR and PPARγ

We first performed molecular docking of melatonin and selected metabolites to the LBD of the AhR in comparison to known natural ligands, such as indirubin, and indole 3-carbinol as well as L-tryptophan, its metabolites, and unrelated indolic or aromatic intermediates of melanogenesis (Appendix A). Then, we performed binding free energy analysis based on the equilibrated last 100 ns MD simulation trajectories from 300 ns MD simulations (Appendix A) to further demonstrate the binding specificity and support the docking results. Molecule docking results (Appendix A) show that melatonin and all the selected metabolites under study had favorable binding to the AhR with docking scores similar or close to those of the natural ligands. Interestingly, the docking scores of melatonin precursor L-tryptophan and of several unrelated intermediates of melanogenesis (L-DOPA, glutathionyl DOPA, 2-S-Cysteinyl-DOPA) were notably less favorable compared to the natural ligands.

The binding free energy results of the AhR with melatonin, 6(OH)melatonin, 5MTT and AFMK compared with natural ligands indirubin and IAA calculated from the equilibrated last 100 ns MD simulation trajectories are summarized in Table 3. Binding free energy results further confirmed that melatonin, 6(OH)melatonin, 5-MTT and AFMK are able to bind the AhR. Melatonin and 5MTT had a stronger binding affinity with the AhR than 6(OH)melatonin and AFMK, which is consistent with experimental observation (Table 2). These compounds show different degrees of binding with the AhR, and different binding energy components, such as van der Waals energy, electrostatics energy and polar solvation energy, were observed between different ligands in complex with the AhR. Different pose orientations and their atomic charge parameter affect the interaction of each ligand with the binding site residues of the AhR, as shown in Figure 4.

Melatonin, 6(OH)melatonin, 5MTT and AFMK were docked into the same binding site of the LBD of the AhR as the natural ligand IAA (Figure 4). The AhR LBD can accommodate a wide range of structurally distinct scaffolds. Whether ligands structurally related to melatonin would have similar/overlapping poses in the promiscuous binding site of the AhR is presently unknown. The four ligands clearly show some similarities; AFMK may be considered an open-chain derivative at the pyrrole moiety compared to the other ligands. We obtained multiple distinct and equivalent poses in the AhR binding site out of which we chose the best scoring pose for each docked ligand. AhR residues surrounding each docked ligand are summarized in Figure 4 and Appendix A. Although the binding site is not relaxed around the bound ligands’ docking, it suggests the following key interactions for each compound: (a) Melatonin may form hydrogen bonding interactions with Gln383 and with the backbone carbonyls of Gly321 and Cys333. Most favorable nonpolar contacts are formed with Phe295, Ile325, Met348, Leu353 and Val381. (b) 5MTT can form a hydrogen bond with the backbone carbonyl of Cys333. Residues Phe295, Y322, His337, Met340, A367 and Val381 contribute through nonpolar interactions. (c) AFMK (N-Acetyl-N-formyl-5-methoxykynuramine) may form a hydrogen bond with Ser336, Ser365 and Gln383. Favorable nonpolar contacts formed with Phe295, Tyr322, Ile325, Cys333, Phe351, Leu353, Ala367 and Val381. (d) 6-(OH)-Melatonin may form a hydrogen bond with Ser320 and Gln383. Nonpolar contacts are contributed by His291, Phe295, Tyr322, Phe324, Ile325, His337, Leu353 and Val381.

Molecular docking was also performed for melatonin and the related metabolites to the LBD of the PPARγ along with the established ligand rosiglitazone (Appendix A). Melatonin and all selected metabolites have favorable binding poses at the PPARγ with docking scores comparable to rosiglitazone. The precursor, L-tryptophan, and an unrelated aromatic precursor to melanogenesis, L-DOPA, have the poorest docking scores. Interestingly, indolic compounds and derivatives of L-DOPA had docking scores relatively similar to the natural ligands, an observation worthy of future investigations.

The binding modes for melatonin, 6(OH)melatonin, 5MTT and AFMK (docked into the same PPARγ binding site as rosiglitazone) are shown in Figure 5. 6PPARγ residues mapping to the binding site region of each ligand are summarized in Figure 5 and Appendix A. Favorable interactions derived from the highest-scoring docked pose of each compound are as follows: (a) Melatonin’s indole ring is sandwiched between the guanidinium moiety of Arg288 and Leu333 while forming a hydrogen bond with the backbone carbonyl of Leu228. The amide may have a hydrogen bond with the carboxyl group of Glu295. Side chains of L228, L330, M329 and A292 contribute to ligand binding through hydrophobic contacts. (b) 5MTT is predicted to form a hydrogen bond with the Glu295 carboxyl group. The ligand is surrounded predominantly by hydrophobic side chains (F226, L228, A292, I326, M329 and L330), while its aliphatic amine is in a partially polar region, near Cys285 and Ser289. (c) AFMK participates in hydrogen bonding with Tyr327 and the backbone carbonyl of Cys285 through its acetamide and with Arg288 through its formamide group. The alkyl chains of Arg288 and Leu228, Cys285, Ile326, Tyr327, Leu330, L333, V339, Ile341 and Met364 are the closest side chains with nonpolar contacts with the ligand. (d) 6-(OH)-Melatonin forms several hydrogen bonding interactions: between 6-OH and Tyr327, 6-methoxy and Lys367, the indole amine and Ser289 and, further, two hydrogen bonds through the acetamide with Arg288 and the backbone carbonyl of Ile326. Nonpolar interactions are formed by the aliphatic alkyl part of the Arg288 side chain and Cys285, Ala292, Ile326, Tyr327, Met329, Leu330, Leu333, Phe363 and Met364. Of the docked ligands, three contain an indole ring with a flexible ‘side chain’. Our docking results suggest multiple possible binding modes for ligands containing an indole with a flexible side chain, each predicting favorable interactions with the PPARγ. Whether there is a single preferred position of the indole moiety in the PPARγ binding site for such ligands or multiple possibilities requires future investigation with molecular dynamic simulations.

## 3. Discussion

In this study, using both receptor binding and functional assays combined with molecular docking and binding free energy analyses from the equilibrated MD simulation trajectories, we show for the first time that melatonin and its metabolites, including 6(OH)melatonin, 5MTT and AFMK, can serve as ligands for the AhR and PPARγ nuclear receptors. These findings represent a significant advancement, since the only recognized receptors for melatonin in mammalian systems were the high-affinity membrane-bound G-protein-coupled MT1 and MT2 receptors [14,15]. Melatonin acts on those membrane-bound receptors at nM or lower concentrations to fulfill its functions as a neurotransmitter, hormone and immunomodulator [13,14,15], while the receptors for its indolic or kynuric metabolites remain to be defined. Furthermore, melatonin and its metabolites exert antioxidative and DNA-protective responses requiring high concentrations of the compounds (1–1000 µM), effects that are MT1- and MT2-independent [21,23,44,45,46,47]. Therefore, concentrations of melatonin and its metabolites at a 10^−3^–10^−6^ M range, which are necessary to activate the AhR or PPARγ, correspond to their concentrations required for antioxidative and radioprotective effects in general or for the phenotypic effects observed in the skin [9,12,23,45,46,47,48,49,50,51]. These concentrations of melatonin or its metabolites are expected to be used for topical preventive or therapeutic purposes on the skin, further signifying the pharmacological utility of the above findings.

The AhR is a basic helix–loop–helix transcription factor, which, after the binding of endogenous or exogenous ligands, translocates to the nucleus, where it dimerizes with the AhR nuclear translocator (ARNT) and binds to the AhR-, dioxin- or xenobiotic-responsive element (AHRE, DRE or XRE) and regulates the transcriptional activity [39,52,53,54]. The AhR has a large and promiscuous LBD that can accommodate diverse synthetic and natural ligands with different structures, including environmental pollutants or endogenous compounds, such as aromatic hydrocarbons, tetrapyrroles, arachidonic acid metabolites, carotenoids and derivatives of tryptophan or indole metabolites [36,37,38,39,40,41,42,55]. In this study, using a Human AhR Reporter Assay System, we demonstrated that melatonin activates the AhR with a potency similar to its natural ligand IAA (Figure 1). This property was also shared by its metabolites, including 6(OH)melatonin, AFMK and 5MTT; however, it has a lower efficiency. The activation of the AhR by melatonin and its metabolites was further documented by the stimulation of the promoter of *CYP1A1*, a gene downstream of the AhR, and stimulation of the AhR protein translocation from the cytoplasm to the nucleus using two independent methods, ImageStream flow cytometry and WB analyses of the cytoplasmic and nuclear fractions (Figure 2). These were further supported by their favorable interactions with the LBD of the AhR, which were predicted through the docking of melatonin and its derivatives, sharing the same ligand-binding pocket with the natural ligand IAA (Figure 4). Also, Glide XP docking scores of melatonin and its metabolites, which docked into the AhR homology model, were comparable to its natural ligands indirubin, indole-3 carbinol and IAA (Appendix A). Thus, both experimental results and modeling predictions indicate that melatonin and its metabolites are ligands for the AhR. The identification of the AhR as the nuclear receptor for melatonin and its metabolites can provide a mechanistic explanation for their cytoprotective effects against ultraviolet radiation or oxidative stress, which are mediated by high concentrations of the ligands and are independent from the MT1 and MT2 expression [23,45,46,48,49,51,56,57,58]. The significance of these findings is enhanced by the reported role of the AhR in the epidermal barrier functions, antioxidative responses and downregulation of pro-inflammatory responses [38,59,60,61,62,63,64,65,66,67].

The PPARγ, while being a master regulator of lipid metabolism in adipose tissue and the liver, exerts additional regulatory functions in several immune cells, various epithelial cells and cancer [68,69,70]. In the skin, it promotes lipid synthesis, a keratinocyte differentiation program and barrier functions [71,72,73,74,75]. Many natural and endogenous products can bind to and activate the transcriptional activity of the PPARγ, including polysaturated fatty acid, arachidonic acid metabolites, synthetic and natural cannabinoids and several synthetic compounds, including glitazones, compounds structurally related to salicylic acid or benzoic acid, or different phytochemical products [68,70,76]. Depending on the structure, these compounds can act at nano-, micro- or millimolar ranges. In the current study, we show for the first time that melatonin and its metabolites can not only bind to the LBD of the PPARγ but also stimulate the PPAR-mediated activation a PPRE-driven promoter at a 10^−4^–10^−3^ M range, concentrations that are several orders of magnitude higher than those of the natural ligands (Figure 3, Table 2). These functional assays were further supported by molecular docking to the LBD of the PPARγ with docking scores comparable to the established ligand rosiglitazone (Appendix A). Melatonin and its tested metabolites fit favorably in the same binding pocket of the PPARγ as rosiglitazone (Figure 5). The established EC_50_ values to the PPARγ are in agreement with the concentrations of 10^−4^–10^−3^ M necessary for the antioxidative and radioprotective functions of melatonin and its metabolites [23,45,46,48,49,51,56,57,58]. Of note, some natural PPARγ agonists, such as compounds of the 5-aminosalicylic acid class, also require a high concentration of the ligand to exert phenotypic activity [70]. Thus, these affinities are clinically relevant.

## 4. Materials and Methods

### 4.1. Chemicals

Melatonin, 2-hydroxymelatonin (2(OH)Mel), 6-hydroxymelatonin (6(OH)Mel), 5-metoxytryptophan (5MTT) and indole acetic acid (IAA) were purchased from Sigma-Aldrich (St. Louis, MO, USA), while *N*^1^-acetyl-*N*^2^-formyl-5-methoxykynuramine (AFMK) was from Cayman Chemical (Ann Arbor, MI, USA). Charcoal-stripped fetal bovine serum (cFBS) was purchased from Atlanta Biologicals (Lawrenceville, GA, USA). Dulbecco’s modified Eagle’s medium (DMEM) with high glucose (4500 mg/L), 1% penicillin-streptomycin solution (10,000 units of penicillin and 10 mg of streptomycin in 1 mL 0.9% NaCl), DMSO, ethanol, HEPES buffer, nonessential amino acids (NEAAs) (100×) and Triton^®^ X-100 were purchased from Sigma-Aldrich (St. Louis, MO, USA). FBS, 0.05% trypsin/0.53 mM EDTA solution, 1 × PBS (pH 7.4) and L-glutamine (200 mM) were supplied by ThermoFisher Scientific (Waltham, MA, USA). Hoechst 33342 Ready Flow Reagent was from ThermoFisher (Waltham, MA, USA, Cat r37165), Fixation Buffer was from Biolegend (San Diego, CA, USA, Cat 420801) and Perm Buffer 3 was from BD Biosciences (Franklin Lakes, NJ, USA, Cat 558050).

### 4.2. Cell Cultures

Immortalized adult human epidermal keratinocytes (HaCaT) were cultured in DMEM supplemented with 5% cFBS and 1% antibiotic-antimycotic solution [48]. Semiconfluent cultures were used for testing the effects of melatonin and metabolites, as detailed below. Briefly, HaCaT cells were washed with DMEM and then treated with DMEM containing 0.5% BSA and 10^−4^ M of either melatonin, 6(OH)Mel, 2(OH)Mel, AFMK, 5MTT or melatonin, 6(OH)Mel, 5MTT and AFMK or 0.1% ethanol (negative control) for 6 and 16 (flow cytometry) or 24 h (WB) and were harvested using trypsin/EDTA solution. After washing in PBS, the cells were submitted for image flow cytometry or extracted for cytoplasmic and nuclear fractions, as described below.

### 4.3. Western Blot Analyses

Cytosolic and nuclear fractions of control and treated HaCaT samples were extracted using a nuclear extraction kit (Active motif, Carlsbad, CA, USA). A Bio-Rad protein Bradford assay kit (BioRad, Hercules, CA, USA) was used for protein quantification. The concentration of protein samples was determined using the Bio-Rad protein Bradford assay kit (BioRad, Hercules, CA, USA). Equal amounts of protein (30 μg per well) were loaded into each well of the gels. Mini-PROTEAN^®^ TGX™gel (BioRad, Hercules, CA, USA) was used for the separation of protein in gels followed by transfer to the PVDF membrane. The membranes were first probed with anti-AhR Antibody (Ab) then stripped and re-probed sequentially with antibodies to Lamin A/C and Tubulin, as shown in Figure 3. The primary antibodies were as follows: mouse monoclonal antibody against AhR (sc-133088) (Santa Cruz Biotechnology, Dallas, TX, USA) diluted by 1:1000, goat polyclonal antibody against Lamin A/C (sc-6215) (Santa Cruz Biotechnology, Dallas, TX, USA) diluted by 1:500 and mouse monoclonal against α-tubulin (DM1A, 62204) (Invitrogen, IL, USA) diluted by 1:2500. The membranes were incubated overnight at 4 °C. The membranes were then incubated for 1 h at room temperature with the HRP-conjugated secondary antibodies using anti-mouse secondary antibody (sc-516102) (Santa Cruz Biotechnology, Dallas, TX, USA) or mouse anti-goat secondary antibody (sc-2354) (Santa Cruz Biotechnology, Dallas, TX, USA) at 1:5000 dilution. Immuno-reactivity was detected using super signal west pico ECL (BioRad, Hercules, CA, USA). The protein bands of interest were identified according to their molecular weights (kDa) published in the manufacturers’ datasheets, determined relative to the precision plus protein™ kaleidoscope™ standards (BioRad, Hercules, CA, USA). The relevant proteins of interest were detected sequentially with the membranes being re-probed with each antibody type, including those for the loading controls. Alpha-tubulin served as the loading control for whole-cell and cytosolic fractions, while Lamin A/C served as the loading control for the nuclear fraction [77,78]. Image J software analyses of blots were performed from 3 independent experiments. To analyze the intensity of relevant bands in the blots, an area was selected around the band at the expected molecular weight (kDa) using Image J software (version V1.54g), and the pixel intensity was measured. Quantitative data were then imported into Microsoft Excel for calculating the percentage intensity of relevant bands compared to the intensities of controls and the ratio of intensities between the nuclear and cytosolic fractions. For the analysis of whole-cell and cytosolic fractions, band intensities were normalized relative to the loading control and to the band intensities seen in control cells (ethanol), and they are presented as a % of the control (mean ± SD). For the analysis of the nuclear to cytosolic ratio, the nuclear-AhR pixel intensities were normalized by dividing the pixel intensity value by that for the Lamin A/C loading control.

### 4.4. Imaging Flow Cytometry

The studies on the melatonin- or metabolite-induced translocation of AhR to the nucleus or mitochondria followed previously described methodology [78,79] with some modifications. Briefly, harvested and washed HaCaT (see above) cells were incubated with Hoechst 33342 Ready Flow Reagent for 50 min, fixed for 20 min in Fixation Buffer and permeabilized in Perm Buffer 3 [77,80]. The cells were then incubated for 20 min at 1:100 in AhR antibody (BD Biosciences, Franklin Lakes, NJ, USA, Cat 565789 AF647). The HaCaT cells were then analyzed via imaging flow cytometry using ImageStream II, as described previously [78], and data were analyzed using IDEAS software version 6.2 (Amnis, Seattle, WA, USA).

### 4.5. Activation of the AhR

To test the effect of melatonin, 6(OH)Mel, 5MTT and AFMK on AhR activation, we used a Human AhR Reporter Assay System (INDIGO Biosciences, State College, PA, USA), as described previously [43]. Briefly, the reporter cells were recovered using the cell recovery medium for 5 h. Then, they were exposed to melatonin, 6(OH)Mel, 5MTT or AFMK or IAA (positive control) in a screening medium for 22 h. Luciferase detection reagent was added after discarding the media and luminescence was measured using a Citation 5 Cell Imaging Multi-Mode Reader (BioTek, Winooski, VT, USA).

### 4.6. Binding to the LBD of the PPARγ

A PPARγ receptor binding assay was performed using the LanthaScreen™ TR-FRET PPARγ Coactivator Assay Kit (Life Technologies Corporation, Grand Island, NY, USA) following the manufacturer’s protocol. Briefly, PPARγ-LBD was added to the compounds, followed by the addition of the antibody (Tb-anti-GST) and a mixture of peptide (Fluorescein-TRAP220/DRIP-2). The reaction mixture was incubated at room temperature for 2 h or overnight. The emission signal at 520 nm was divided by the emission signal at 495 nm for the TR-FRET ratio using Synergy neo2 (BioTek, Winooski, VT, USA).

### 4.7. Luciferase (LUC) Assays

The AhR-mediated transcriptional activity was also tested using CYP1A1 promoter in a pGL4.23-1A1 construct coupled to the LUC (provided by Dr. Sutter, University of Memphis). HaCaT cells were transfected with pGL4.23-1A1-LUC constructs and co-transfected with pRL (Renilla Luciferase Control Reporter Vectors) in Lipofectamine 3000 (Invitrogen, Waltham, MA, USA), followed by treatment of the compounds for 24 h. Luciferase activity was determined by measuring luminescence using the Dual-Luciferase^®^ Reporter Assay System (Promega Corporation, Madison, WI, USA), following the manufacturer’s protocol with Citation 5.

The transcriptional activation of the PPRE promoter was tested using the J_3_TKpGL3 reporter plasmid, which contains a luciferase gene driven by the PPAR-responsive element (PPRE), which consists of three copies of the J site from the apo-AII gene promoter [81]. The activation of PPAR elements was evaluated in melanoma cells following stimulation with a Firefly–Renilla Dual-Luciferase Reporter System (Promega, Madison, WI, USA) [82,83]. Experiments were carried out using human melanoma line WM239A and A375P (Trust Functional Genomics Cell Bank, London, UK) cells cultured in RPMI culture medium (ThermoFisher Scientific, Waltham, MA, USA) supplemented with 10% fetal bovine serum (EURx, Gdansk, Poland) at 37 °C, with a 5% CO_2_ content [81]. Four hours after transfection, cells were treated for 40 h with 10^−4^ M melatonin, 5(OH)tryptophan, 6(OH)melatonin or AFMK or 2 × 10^−5^ M pioglitazone or fenofibrate. Luciferase activity was measured using a commercial kit (Luciferase Assay System, Promega) according to the instructions of the supplier and assessed in a chemiluminometer (Synergy HT; Bio-Tek, Winooski, VT, USA)) using the Dual-Reporter Luciferase Assay System (Promega, Madison, WI, USA).

### 4.8. Molecular Docking of Ligands Targeting AhR and PPARγ

To predict the binding of melatonin and its metabolites to the AhR and PPARγ, molecular docking was performed using Glide in Extra Precision (XP) implemented in Schrödinger [84] (version 2016-1) by OTAVA LTD (Vaughan, Ontario, L4K 0C3, Canada), as described previously [43,78].

The human AhR structure is a homology model described previously [43]. The PPARγ was obtained from Protein Data Bank (PDBID:3TY0). It was optimized with an OPLS2005 force field with convergence of heavy atoms to RMSD 0.3 Å. Receptor grids were constructed as described previously [43]. Three-dimensional structures of the ligands were prepared using the LigPrep utility of Schrödinger with OPLS2005 force field and ionization states generated at a pH of 7.0 [78]. Three-dimensional receptor–ligand interactions for each studied receptor were visualized using PyMol software (https://pymol.org/2/ accessed on 13 April 2022) and 2D interaction maps were generated in Maestro version 12.4 (Schrödinger software, https://www.schrodinger.com/ accessed on 13 April 2022).

### 4.9. Molecular Dynamics Simulation and Binding Free Energy Analyses of AhR–Ligands Complex

Melatonin, 6(OH)melatonin, 5MTT and AFMK in complex with the AhR were selected for MD simulation compared with natural ligands indirubin and IAA. A total of 300 ns MD simulations were conducted using the Amber 14 MD simulation package (https://ambermd.org/ (accessed on March 3, 2020)) to determine the binding mode and binding free energy. An AMBER force field was used for the simulated systems. MD simulation protocols are the same as those in our previous studies [43,78,85].

To determine the simulated systems’ equilibration tendencies and its convergence, the root mean square deviation (RMSD) of protein backbone atoms over time was analyzed.

The binding free energy was determined with the Molecular Mechanics Poisson-Boltzmann Surface Area (MM-PBSA) method, as described in our previous studies [43,78,85].

### 4.10. Statistical Analyses

Experimental data are expressed as mean ± standard error or standard deviation, when indicated, of at least three separate assays (*n* ≥ 3). The experiments were repeated at least two times. The significance of individual treatment groups in comparison to the control group (solvent) was evaluated using a one-way ANOVA or the Student *t*-test, where appropriate. * *p* < 0.05, ** *p* < 0.01, *** *p* < 0.001 and **** *p* < 0.0001 for all experiments, using Prism, version 10 GraphPad Software Inc., (La Jolla, CA, USA).

## 5. Conclusions

Our functional and transcription data together with our results from molecular docking support our conclusion that melatonin and its metabolites act as ligands for the AhR and PPARγ. This opens exciting areas for research on their mechanisms of action and coupling to different signal transduction pathways that would be context- and cell type-dependent. Relatively high EC_50_ values of evaluated compounds towards the AhR and PPARγ are consistent with the requirement for similarly high concentrations of the ligands to exert cytoprotective, photo/radioprotective, antimelanogenic and antioxidative effects. The significance of these findings is enhanced by the ability of many peripheral organs to synthesize and metabolize melatonin, independently of the pineal gland, leading to very high local concentrations of melatonin exceeding the ones observed in serum. In these organs, including skin, melatonin and its metabolites would act on the AhR and PPARγ in intra-, auto- or paracrine fashions. The therapeutic utility of pharmacological doses of the compounds is also envisioned, since they show absent or relatively low toxicity.

## Figures and Tables

**Figure 1 ijms-24-15496-f001:**
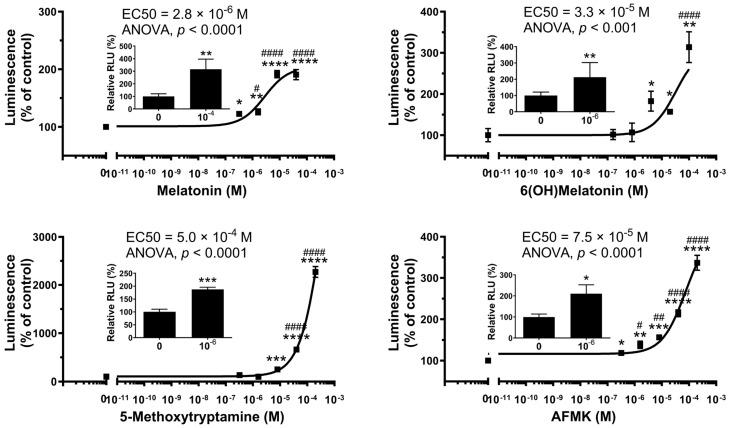
Melatonin and derivatives act on the AhR. A cell-based Human AhR Reporter Assay System kit was used. Inserts: the transcriptional stimulation of the *CYP1A1* promoter containing AhR-RE. The HaCaT keratinocytes were transfected with the construct 48 h after stimulation with the ligand, as described in the Section 4. Luminescence was analyzed using the Dual-Luciferase Reporter Assay System. The values represent means ± SE (n ≥ 3). Data were analyzed using a one-way ANOVA (Dunnett’s multiple comparison test) where # *p* < 0.5, ## *p* < 0.01 and #### *p* < 0.0001 and a Student *t*-test where * *p* < 0.05, ** *p* < 0.01, *** *p* < 0.001 and **** *p* < 0.0001.

**Figure 2 ijms-24-15496-f002:**
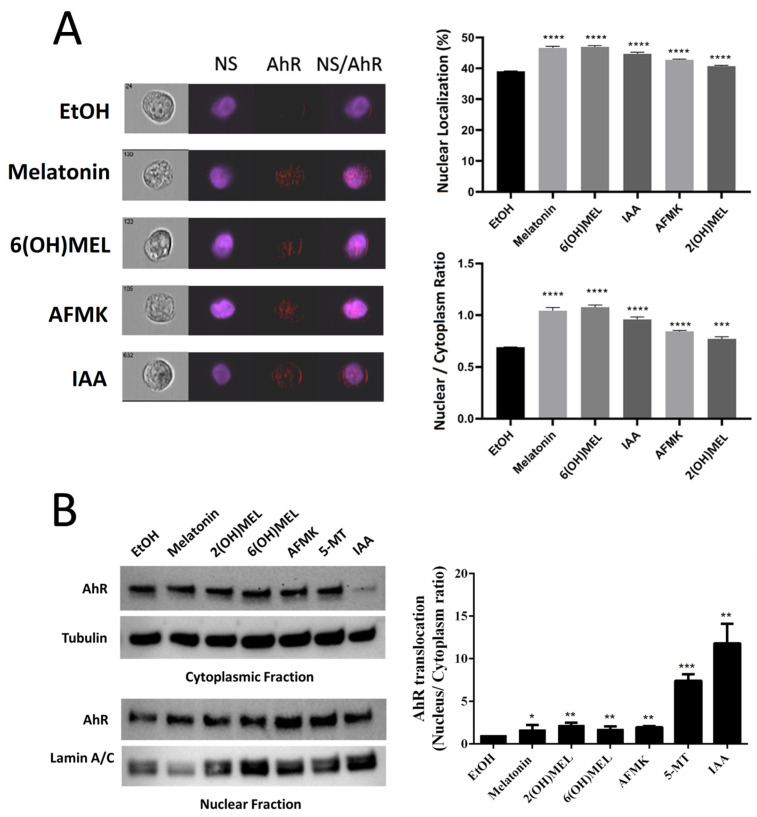
Ligand-induced translocation of AhR to the nucleus by melatonin and its metabolites. (**A**) Image flow cytometry analysis; (**B**) WB analysis of cytoplasmic and nuclear fractions. ETOH: vehicle, 5MTT: 5-methoxytryptamine, IAA: indole acetic acid, 2(OH)MEL: 2-hydroxymelatonin, 6(OH)MEL: 6-hydroxymelatonin, AFMK; **** *p* < 0.0001; *** *p* < 0.001; NS: *p* > 0.05. The cells were treated with 10^−4^ M of either melatonin, 6(OH)Mel, 2(OH)Mel, AFMK and 5MTT or melatonin, 6(OH)Mel, 5MTT and AFMK or 0.1% ethanol (negative control) for 6 (flow cytometry) or 24 h (WB) and were harvested using trypsin/EDTA solution. After being washed in PBS, the cells were submitted to image flow cytometry or extracted for cytoplasmic and nuclear fractions as described in the Section 4. (**A**) Left panel: cytometry images of individual cells showing AhR localization in cytoplasm or nucleus following treatments with the compounds. NS: stain for nuclei; AhR: stain for AhR antigen; NS/AhR: combined stain for both nuclei and AhR antigen. Right panels: percentage of AhR staining in the nucleus is in the upper panel. Ratios of nuclear (co-localization with Hoechst) vs. cytoplasmic localization of AhR determined following analysis of individual cells (n = 515 to 2339) is in the lower panel. Bar graphs represent the quantitative analysis of images acquired by cytometry ± SE. (**B**) The representative WB is on the left, quantification of the signal ratios is on the right. The values represent means ± SD from three independent experiments with values being statistically significant: * *p* < 0.05; ** *p* < 0.01; *** *p* < 0.001.

**Figure 3 ijms-24-15496-f003:**
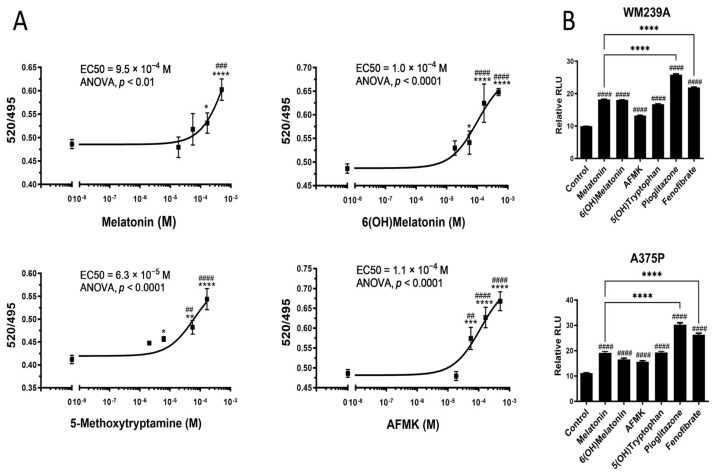
Activation of PPAR by melatonin and its derivatives. LanthaScreen™ TR-FRET Coactivator Assay Kit was used for analyzing ligand binding to PPARγ (**A**). The transcriptional stimulation of the PPAR-RE was determined using a Dual-Luciferase Reporter Assay System (**B**). Data were analyzed using a one-way ANOVA (Dunnett’s multiple comparison test) where ## *p* < 0.01, ### *p* < 0.001 and #### *p* < 0.0001 and a Student *t*-Test where * *p* < 0.05, ** *p* < 0.01, *** *p* < 0.001 and **** *p* < 0.0001 in (**A**) and a one-way ANOVA (Dunnett’s multiple comparison test) where #### *p* < 0.0001 comparing the control and **** *p* < 0.0001 comparing melatonin as a positive control in (**B**). The values represent means ± SE (n ≥ 3) (**A**) or (n = 12) (**B**).

**Figure 4 ijms-24-15496-f004:**
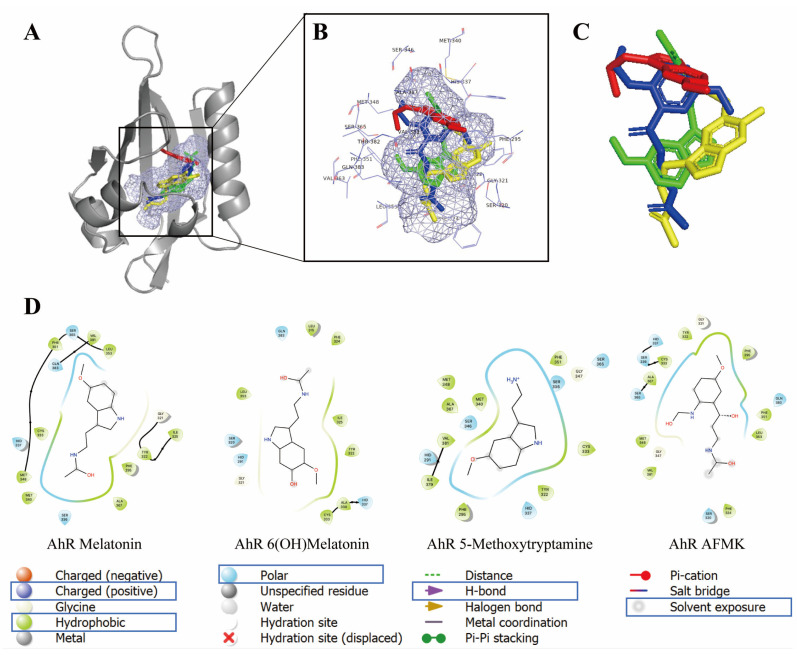
Binding modes for the selected four melatonin derivatives in ligand-binding domain (LBD) of AhR (full view in (**A**) and zoomed in (**B**,**C**)): melatonin (green), 6(OH)melatonin (yellow), 5MTT (red) and AFMK (blue). In (**A**,**B**) the ligand-binding pocket is shown in light blue mesh. In (**A**) the AhR receptor is shown as a cartoon in gray. In (**B**) residues related to the binding pocket are in line representation. (**C**) illustrates docked poses (ligands only). In (**D**) two-dimensional interaction map of melatonin, 6(OH)melatonin, 5MTT and AFMK is shown with the AhR receptor. Images were generated with Maestro (v12.4). Amino acids with positive charge are blue; hydrophobic ones are green; polar ones are light blue. Hydrogen bonds are labeled in purple.

**Figure 5 ijms-24-15496-f005:**
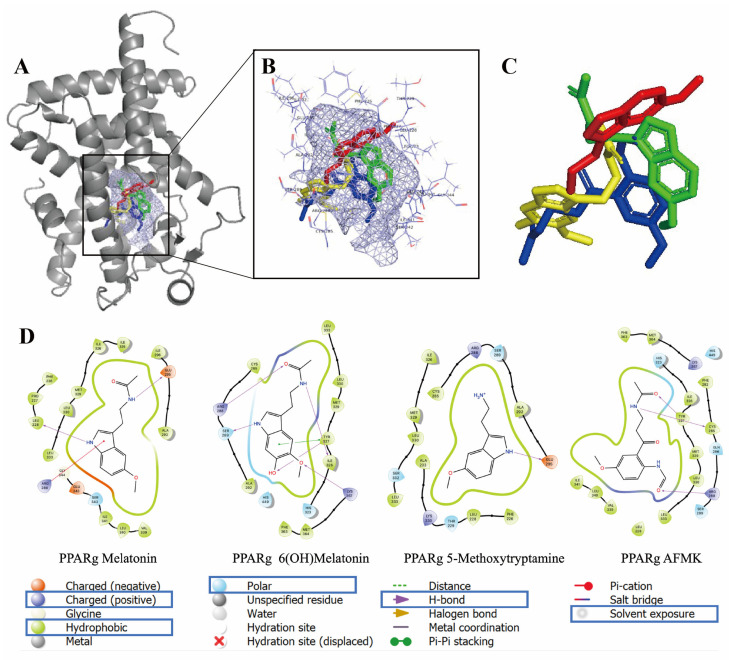
Binding modes for the selected four melatonin derivatives in ligand-binding domain (LBD) of PPARg (full view in (**A**) and zoomed in (**B**,**C**)): rosiglitazone (brown), melatonin (green), 6(OH)melatonin (yellow), 5MTT (red) and AFMK (blue). In (**A**,**B**) the ligand-binding pocket is shown in the light blue mesh. The PPARγ receptor is shown as a cartoon in gray. In (**B**) residues related to the binding pocket are in line representation. (**C**) illustrates docked poses (ligands only). In (**D**) two-dimensional interaction map of melatonin, 6(OH)melatonin, 5MTT and AFMK with the PPARγ receptor is shown. Images were generated with Maestro (v12.4). Amino acids with positive charge are blue; hydrophobic ones are green; polar ones are light blue. Hydrogen bonds are labeled in purple.

**Table 1 ijms-24-15496-t001:** Structures of selected compounds under study.

Name	Structure
*L*-tryptophan	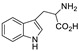
*N*-acetylserotonin	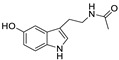
Melatonin	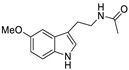
6-hydroxymelatonin	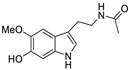
2-hydroxymelatonin	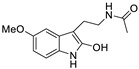
*N*-acetyl-*N*-formyl-5-methoxykynuramine	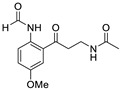
Indole-3-carbinol	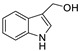
Indirubin	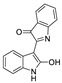
Pioglitazone	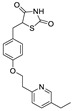
Rosiglitazone	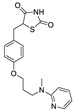
Fenofibrate	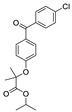

**Table 2 ijms-24-15496-t002:** EC50 values of the binding of melatonin and its metabolites to the AhR and PPARγ ligand-binding domain. ND, not determined. The data represent two independent dose responses for each compound using 3 to 4 replicas per assay.

Compounds	EC_50_ (M)	
	AhR	PPARγ
Melatonin	7 × 10^−6^ ± 7 × 10^−6^ (*n* = 2)	1.2 × 10^−3^ ± 0.3 × 10^−3^ (*n* = 2)
6-Hydroxymelatonin	3.1 × 10^−5^ ± 0.3 × 10^−5^ (*n* = 2)	3.7 × 10^−3^ ± 5.0 × 10^−3^ (*n* = 2)
AFMK	4.4 × 10^−5^ ± 4.5 × 10^−5^ (*n* = 2)	2.8 × 10^−4^ ± 2.4 × 10^−4^ (*n* = 2)
5-MTT	2.9 × 10^−4^ ± 3.0 × 10^−4^ (*n* = 2)	3 × 10^−5^ ± 4 × 10^−5^ (*n* = 2)
Indole Acetic Acid	2 × 10^−6^ ± 2 × 10^−6^ (*n* = 2)	ND
Rosiglitazone	ND	1 × 10^−8^ ± 0.6 × 10^−8^ (*n* = 2)

**Table 3 ijms-24-15496-t003:** Binding free energy of natural ligands of indirubin and indole acetic acid with AhR receptor and selected four melatonin derivatives with the AhR receptor (Kcal/mol).

	Indirubin	Indole Acetic Acid	Melatonin	6(OH)Melatonin	5-Methoxytryptamine	AFMK
ΔE_vdW_	−37.78 ± 0.66	−24.15 ± 3.17	−39.11± 2.56	−35.26 ± 3.27	−17.50 ± 12.55	−39.63 ± 2.90
ΔE_electrostatic_	−122.41 ± 1.61	−172.03 ± 1.62	−24.46 ± 4.08	−17.07 ± 6.05	−21.23 ± 24.08	−27.93 ± 2.90
ΔG_nonpolar-solvation_	−4.42 ± 0.02	−3.54 ± 0.02	−4.55 ± 0.09	−4.56 ± 0.13	−2.86 ± 1.24	−4.92 ± 0.13
ΔG_polar-solvation_	138.62 ± 2.25	169.65 ± 0.93	39.82 ± 3.76	34.71 ± 5.52	12.52 ± 24.86	46.48 ± 4.65
ΔTS	−17.43 ± 9.61	−17.40 ± 9.59	−22.70 ± 11.28	−21.42 ± 10.50	−17.94 ± 10.48	−22.70 ± 10.09
**ΔG_binding_**	**−8.43 ± 3.54**	**−12.60 ± 3.63**	**−5.60 ± 5.58**	**−0.77 ± 5.85**	**−11.13 ± 9.02**	**−3.92 ± 6.66**

All values in this table were expressed in term of Kcal/mol. ΔE_vdW_, van der Waals energy; ΔE_electrostatic_, electrostatic energy; ΔG_nonpolar-solvation_, nonpolar solvation energy; ΔG_polar-solvation_, polar solvation energy; ΔTS, energy contributed from solute entropy; ΔG_bingding_, binding free energy for the complex.

## Data Availability

Not applicable.

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
