# Peer review of "Melatonin and Its Metabolites Can Serve as Agonists on the Aryl Hydrocarbon Receptor and Peroxisome Proliferator-Activated Receptor Gamma"

_ijms, 2023, doi:10.3390/ijms242015496_

Round 1
Reviewer 1 Report
Major comments:
1. EC50 values of AhR in the other of 1 to 100 µM for melatonin are higher than natural levels and far higher than the EC50 values of MT1 and MT2. The same applies for PPARg. Please discuss in more detail that the described finding not only artifacts of superphysiological melatonin concentrations.
2. Table 2: There should be at least 3 repeats of this experiment.
3. The docking experiments are not convincing, please provide more evidence. For example, molecular dynamics simulations.
Minor comments:
1. Please define all abbreviations at their first time use and applied them then consistently (this applies also to the Abstract).
2. Gene and mRNA name abbreviations should be in italic.
3. Please harmonize the font size of all figures, to have them readable.
Author Response
We thank the reviewer for his/her time and effort to improve our presentation.
- EC50 values of AhR in the other of 1 to 100 µM for melatonin are higher than natural levels and far higher than the EC50 values of MT1 and MT2. The same applies for PPARg. Please discuss in more detail that the described finding not only artifacts of superphysiological melatonin concentrations.
Reply
The reviewer is correct the values are high. However, these are expected for the requirements of high concentrations of melatonin and metabolites to exert photoprotective, antioxidative and antimelanogenic effects in the skin. These concentrations are at 10 to 1000 µM, as reported by us and by many other investigators. In addition, these effects that required high ligand concentrations, were independent of MT1 and MT2 receptors. Furthermore, in many peripheral organs, including skin, melatonin is both synthesized from tryptophan and metabolized This leads to very high intracellular or intra-tissue concentrations of these compounds, consistent with present EC50 values,
These issues are discussed in depth in the revised manuscript in the introduction, discussion and conclusion sections as requested by the reviewer
- Table 2: There should be at least 3 repeats of this experiment.
Reply
The interaction of melatonin and metabolites with the nuclear receptors is documented by independent assays: using commercially available kits (table and dose responses), transcriptional activation of the downstream targets, gene reporter assays, translocation into nucleus both by WB and image flow cytometry and molecular modeling. Therefore, there are more than 3 independent repeats.
In compliance to the critique we have provided a more detailed discussion of the data presented in table 2. For example, on page 4: “These values representing two independent experiments for each compound performed in 3 or 4 replicates are shown in table 2.” And on page 8 “The EC50 values in table 2 are from two independent experiments with 3 to 4 replicates per assay.”
- The docking experiments are not convincing, please provide more evidence. For example, molecular dynamics simulations.
Reply
Molecular dynamic simulation are now provided in the supplemental file. In addition, we added the binding free energy Table 3 calculated from MD simulation trajectories. We thank the reviewer for this request. In addition, data from multiple I ndependent functional assays are all consistent with the conclusions obtained from the docking experiments
Minor comments:
- Please define all abbreviations at their first time use and applied them then consistently (this applies also to the Abstract).
Reply
This has been done, both in the abstract and in text.
- Gene and mRNA name abbreviations should be in italic.
Reply
This has been done in the text.
- Please harmonize the font size of all figures, to have them readable.
Reply
The font size was corrected to make them more readable.
In summary, we thank the reviewer for the critique that has allowed improving our presentation.
We hope that this significantly revised version of the manuscript is now acceptable for publication
Reviewer 2 Report
The major issue I have with this study isthe concentrations of melatonin and metabolites used. The best EC50 values reported are >100 fold above known ligands for AhR and PPARg and EC50 for MT1/2 receptors, and >1000 fold above reported physiological melatonin concentrations. How does the reported very modest activation of AhR (melatonin only increased the reporter maximum 2 fold) have any physiological relevance? Under what conditions are cells in the body exposed to these concentrations? As nM neurotransmitters I would have thought that these concentrations would be potentially neurotoxic. Is this simply structures with some physical homology with real ligands being able to exert a similar activity when used at very high concentrations? The modelling data likewise.
In figure 2, I cannot see the AhR in the imaging shown. How sensitive is the antibody and detection? If this is insensitive as suggested by the imaging, then how reliable are the very small changes in nuclear localisation of AhR reported and the calculated nuclear cytoplasmic rations? The immunoblotting data is difficult to interpret. Apart from IAA where there is almost no AhR, which is not commensurately increased in the nuclear fraction, and the loading of the nuclear fractions unequal, how reliable is this data?
ok
Author Response
The major issue I have with this study is the concentrations of melatonin and metabolites used. The best EC50 values reported are >100 fold above known ligands for AhR and PPARg and EC50 for MT1/2 receptors, and >1000 fold above reported physiological melatonin concentrations. How does the reported very modest activation of AhR (melatonin only increased the reporter maximum 2 fold) have any physiological relevance? Under what conditions are cells in the body exposed to these concentrations? As nM neurotransmitters I would have thought that these concentrations would be potentially neurotoxic. Is this simply structures with some physical homology with real ligands being able to exert a similar activity when used at very high concentrations? The modelling data likewise.
Reply
We understand the reviewer concern, which would be valid for action on membrane bound MT1 and MT2 receptors and for neurotransmitter role of melatonin. However, these high values are consistent with our and other studies performed on skin cells studying antioxidative, antimelanogenic (inhibition if melanin pigmentation) and photoprotective properties of melatonin and its metabolites. The required concentrations of the active molecules were from 10 to 1,000 µM, being in the same range as reported in this paper. In addition, we envision topical application of melatonin in the skin for therapeutic or preventive purposes that requires suprapharmacological concentrations to exert desirable phenotypic effects.
These issues are now explained and discussed in corresponding sections of the extensively revised manuscript.
In addition, skin and many peripheral tissues synthesize melatonin from the tryptophan de novo, making its supply independent from the pineal gland. These local concentration can be very high and 10,000 times or more higher than reported for serum levels. Being also produced intracellularly, they will act on site on corresponding nuclear receptors in a manner independent of membrane bound receptors. Therefore, these values are physiologically relevant, taking into consideration, intra-, auto- or paracrine mechanisms of action. The modelling data is supported by functional, transcriptional, and receptor assays.
We have now extensively discussed these issues in the Discussion and Conclusions sections. In addition, we provided the binding free energy Table calculated from MD simulation trajectories. Also, we have added molecular dynamic simulation in the supplemental file that further support molecular modeling.
In figure 2, I cannot see the AhR in the imaging shown. How sensitive is the antibody and detection? If this is insensitive as suggested by the imaging, then how reliable are the very small changes in nuclear localisation of AhR reported and the calculated nuclear cytoplasmic rations?
Reply
The instrument used for acquisition of these data was a three laser ImageStream Mark II with a capability to detect eight fluorescent parameters from two CCD detectors. The instrument is very sensitive for imaging single cells in a stream and capturing fluorescence in high detail; essentially combining the capabilities of a fluorescent microscope and a flow cytometer. However, because acquisition of data is performed of cells in s stream, the sharpness is less than a fluorescent/confocal microscope. The major advantage is the ability for unbiased data analysis. In Figure 2, AHR is stained with an antibody that is conjugated to Alexa647 (A647) and data is collected after excitation at 642 nm laser. Hence, AHR is red in the images. Intentionally, we did not use high gain while collecting the data to avoid artificial brightening of the staining. This approach sometimes sacrifices sensitivity for specificity. The software for data analysis, IDEAS, essentially performs unsupervised analysis once the parameters are defined. The software allows for the setting “masks” to define regions within cells based on defined criteria. For the analysis in Figure 2, we set a mask to define nucleus based on Hoechst nuclear staining (purple). The region outside the nucleus is defined as the cytoplasm. We are able obtain the level of AHR (red fluorescence intensity) within the nucleus and outside the nucleus for 100s of individual cells without any selection bias. In fact, the software prevents any selection bias. From the data, we calculate nuclear to cytoplasm ratio.
The images show brightfield, nuclear staining (NS – Hoechst), AHR and merged (NS/AHR). In the merged image, one can see red staining within the nucleus. For each treatment we analyze the data from close to a 1000 individual cells without any selection bias, hence allowing of high power for sensitive assessment for changes in expression and/or localization. Thus, small changes are accurate.
These issues are now discussed in the section where the corresponding data is presented. In addition, we have exchanged the figure, which is now more readable.
The immunoblotting data is difficult to interpret. Apart from IAA where there is almost no AhR, which is not commensurately increased in the nuclear fraction, and the loading of the nuclear fractions unequal, how reliable is this data?
Reply
The immunoblotting represents data from three independent experiments. The data are statistically significant and consistent with other assays, therefore, highly reliable. This is now indicated in the text, in the corresponding Results section.
In addition, the relevant proteins of the nuclear and cytoplasmic fractions were detected sequentially with the membranes being re-probed with each antibody type, including those for the loading controls. Alpha-tubulin served as the loading control for whole cell and cytosolic fractions, while Lamin A/C served as the loading control for the nuclear fraction. Image J software analyses of blots were performed from 3 independent experiments. To analyze the intensity of relevant bands in the blots, an area was selected around the band at the expected molecular weight (kDa) using Image J software, and the pixel intensity measured. Quantitative data were then imported into Micro-soft Excel for calculating the percentage intensity of relevant bands compared to the intensities of controls and the ratio of intensities between the nuclear and cytosolic fractions. For the analysis of whole cell and cytosolic fractions, band intensities measured by Image J were normalized relative to the loading control and to the band intensities seen in control cells (ethanol), and are presented as % of control (mean ± SD). For the analysis of the nuclear to cytosolic ratio, the nuclear-AhR pixel intensities were normalized by dividing the pixel intensity value by that for the Lamin A/C loading control.
We thank the reviewer for critique and are providing more detailed description of the methodology and quantification in the corresponding section of the paper.
In summary, we thank you for the critique that has improved our presentation and hope that this significantly revised version of the manuscript is now acceptable for publication.
Round 2
Reviewer 1 Report
none
none
Author Response
There are no comments during this review cycle for the reviewer.
We again thank the reviewer for the critique, and made additional improvements on the revised version 2 of the manuscript
Reviewer 2 Report
I have read the authors responses and the changes to the manuscript. The original problem remains. Justifying the supraphysiological concentrations of melatonin and metabolites on the basis that in other papers you have found similar supraphysiological concentrations can influence other cellular functions is a circular argument. In your response you have claimed that these concentrations can be obtained in within cells, but I failed to find where this was referenced. Your idea that these concentrations can be provided by topical application changes the concept of the manuscript. At present, on reading the manuscript it would appear that you are making the claim that the effects observed are normal physiological effects of melatonin. Unless you can provide evidence that these concentrations can be reached in vivo, the focus of the manuscript should be altered to emphasise the potential therapeutic benefits of application of these supraphysiological melatonin doses.
I understand the use of Image Stream imaging but in this case I don't think it is the best option for your application. The very small cytoplasmic cross-section makes changes in cytoplasmic protein levels exceedingly difficult to measure. The immunoblots are unchanged from the initial submission.
Author Response
I have read the authors responses and the changes to the manuscript. The original problem remains. Justifying the supraphysiological concentrations of melatonin and metabolites on the basis that in other papers you have found similar supraphysiological concentrations can influence other cellular functions is a circular argument. In your response you have claimed that these concentrations can be obtained in within cells, but I failed to find where this was referenced. Your idea that these concentrations can be provided by topical application changes the concept of the manuscript. At present, on reading the manuscript it would appear that you are making the claim that the effects observed are normal physiological effects of melatonin. Unless you can provide evidence that these concentrations can be reached in vivo, the focus of the manuscript should be altered to emphasise the potential therapeutic benefits of application of these supraphysiological melatonin doses.
Reply
We appreciate the reviewer comment that apply to systemic actions of melatonin which are driven by its production by pineal gland, and its nM concentrations in the serum. However, the main message of this paper is to emphasize melatonin and its metabolites action in the peripheral organ in which they are produced. Our position is that intra-, auto- and para-crine mechanism of action are also physiological at the local level. The capability of producing melatonin locally is described in the introduction and discussion with many references cited.
To address the point of “physiological” and “unphysiological”, we have removed referrals to physiologically relevant and used instead term clinically relevant, or indicated that they would act in intra-, auto- or paracrine-fashions, which would require high local concentrations of the compounds.
Accordingly: corresponding revisions are included in last sentence of the abstract (lines 47-49); in the introduction (lines 62-64; lines 73-74). Also, we modified the discussion to clarify these points (lines 329-342; lines 386-395). We also revised the Conclusions, see lines 570-579.
Also, the therapeutic utility of pharmacological doses of these compounds is re-emphasized, see lines 49-51 in the Abstract and lines 575-577 in Conclusions
I understand the use of Image Stream imaging but in this case, I don't think it is the best option for your application. The very small cytoplasmic cross-section makes changes in cytoplasmic protein levels exceedingly difficult to measure. The immunoblots are unchanged from the initial submission.
Reply
Image flow: we provide additional clarifications to guide unprepared reader, see lines 138-145. The reader can enhance the size of the image, since this is an electronic journal. However, if still necessary we can select the best picture for translocation and show it in the supplement.
WB: We do not understand the critique. On the left there is representative WB on the right quantification from 3 experiments. In fact all and unedited images of WB were provided as supplement per journal regulations. However, to provide better clarification we have added new information and clarifications on line 147-155 and 176-179.
I hope that the revised manuscript is now acceptable for publication in the IJMS, and we hope that the reviewer is not offended by our standing that local production of the ligand to act locally at high concentrations is also physiological from the intrinsic mechanism of action at the periphery. In fact, in compliance we removed statements on physiological functions leaving this point for the reader’s judgement. We have also emphasized therapeutic utilities of pharmacological doses of melatonin and metabolites, since these are non-toxic or show relatively low toxicity.
In addition, the previous changes were accepted to make the final version easy to follow.
Round 3
Reviewer 2 Report
The authors have addressed in part my major criticisms of the revised manuscript. The effects being reported are modest and require supraphysiological concentrations to melatonin and metabolites. The authors do not directly claim and provide no evidence that these can be reached physiologically, although this is possible as a pharmaceutical approach. I remain sceptical of the contribution of melatonin even in a pharmaceutical setting to modulate these activities in a meaningful manner, but the actual results are for the most part sound. I still find the nuclear localisation data very weak, the immunoblots all appear very similar in levels so the changes for most metabolites are very modest, only IAA cause a significant change and that appears to be due to loss of the cytoplasmic fraction of the protein, there does not appear to be a compensatory increase in the nuclear level. However, on the basis of the changes this is acceptable for publication.